# Care navigation addresses issues of tele-mental health acceptability and uptake in rural and remote Australian communities

**Olivia J. Fisher**[1,2‡]*, **Kelly McGrath**[1,3‡], **Caroline Grogan**[1], **Wendell Cockshaw**[1,2], **Chez Leggatt-Cook**[4]

1 Health Services Research, Wesley Research Institute, Brisbane, Queensland, Australia, 2 Faculty of Health, Charles Darwin University, Darwin, Northern Territory, Australia, 3 Isaac Navicare Hub, Wesley Research Institute, Moranbah, Queensland, Australia, 4 Family and Disability Services, UnitingCare Queensland, Brisbane, Queensland, Australia

‡ OJF and KM are contributed equally to this work and are co-lead authors.
* olivia.fisher@wesleyresearch.org.au

## Abstract

### Introduction

People living in rural and remote areas face substantial barriers to accessing timely and appropriate mental health services. In the Bowen Basin region of Queensland, Australia, barriers include: limited local providers, long waiting lists, unreliable telecommunication, and reluctance to trial telehealth. Isaac Navicare is a new, community co-designed care navigation service which addresses these barriers by coupling care navigation with supported telehealth, and referrals to mental health providers and other supports. We aimed to understand the reach and effectiveness of Isaac Navicare in improving access to mental health services and address an evidence gap on strategies for improving telehealth acceptability.

### Methods

This mixed-methods implementation science evaluation used the RE-AIM Framework. It involved a client database review, survey and semi-structured interviews with service users during the 12-month pilot from November 2021.

### Results

197 clients (128 adults, 69 minors) were referred to Navicare during the pilot. Half of adult clients were unemployed, meaning referral options were limited to low-cost or bulk-billed services. Participants described Navicare as supportive and effective in helping to access timely and appropriate mental health supports. Most clients who expressed a treatment modality preference selected face-to-face (n = 111, 85.4%), however most referrals were for telehealth (n = 103, 66.0%) due to a lack of suitable alternatives. The rapport and trust developed with the care navigator was critical for increasing willingness to trial telehealth. Barriers to telehealth included privacy issues, technical difficulties, unreliable internet/phone, and perceived difficulties developing therapeutic rapport. The supported telehealth

**Data Availability Statement:** Access to data is restricted for ethical reasons as imposed by the UnitingCare Queensland Human Research Ethics

Committee, Reference: Fisher_20220926. The data contain potentially sensitive and identifiable information. Data requests can be made via the UnitingCare Queensland Human Research Ethics Committee at hrec@ucareqld.com.au.

**Funding:** Service Delivery: Since 2020, Mitsubishi Development Pty Ltd has funded the service development and delivery of Isaac Navicare. Additional service delivery funding was received from BHP in 2022 to employ an additional care navigator. Research: This study was funded by a donation by Mitsubishi Development Pty Ltd of $25,000.00. The service delivery and research funding supported the salary of authors KM and CG. The funder did not have any additional role in the study design, data collection and analysis, decision to publish, or preparation of the manuscript. The specific role of these authors is outlined in the 'author contributions' section.

**Competing interests:** The authors OF, CG and KM acknowledge a potential conflict of interest due to their roles in the design (OF) or service delivery (KM and CG) of Isaac Navicare. To minimise any impact of this potential bias, CL-C was invited to be a member of the research team external to the project with no conflict of interest to declare. CL-C reviewed a selection of transcripts along with the quantitative data to ensure the authenticity of the results presented, and the accurate interpretation of these data. CL-C also contributed to the development of the manuscript. The funder, Mitsubishi Development Pty Ltd. has no ownership of intellectual property, patents, products in development, marketed products, or access to client data collected during the normal operation of Isaac Navicare. The funder also has no access to, or ownership of research data collected during this study. This commercial affiliation does not alter our adherence to PLOS ONE policies on sharing data and materials.

site was under-utilised. The majority (88.3%, n = 182) of referrals to Navicare were from local health or community service providers or schools.

## Discussion

Coupling supportive, individualised care navigation with tele-mental health provider options resulted in increased uptake and acceptance of telehealth. Many barriers could be addressed through better preparation of clients and improving promotion and uptake of the supported telehealth site.

## Conclusion

Attitudes towards telehealth have changed during the COVID-19 pandemic, however although the need exists, barriers remain to uptake. Telehealth alone is not enough. Coupling telehealth with other supports such as care navigation improves acceptance and uptake.

## Introduction

Mental health systems are notoriously difficult to navigate, particularly for those living outside metropolitan areas. In Australia, although the prevalence of mental illness is around 20% in any 12-month period, regardless of whether a person lives in a metropolitan, regional, rural or remote area [1], the likelihood of severe illness and potentially preventable hospitalisations increases with remoteness [1–3] due to lack of access to appropriate and timely supports [2]. Much has been written about the opportunities that tele-mental health offers. Telehealth alone, however, is not enough to address systemic disadvantages such as limited access to reliable phone and internet services, pervasive stigma, negative attitudes towards mental health and help-seeking, and reluctance to engage in telehealth modalities [4–8]. Reliance on general practitioners to navigate help-seekers through the mental health system has not been effective, with more than one-quarter of help-seekers who consulted a general practitioner in 2020–21 reporting an inability to access appropriate mental health supports [1]. Those living in rural and remote areas may have reduced access to reliable internet and technological infrastructure, and some help-seekers, particularly those from disadvantaged groups, may lack necessary skills, confidence or equipment [9]. Tailored solutions are required to improve the uptake and acceptability of tele-mental health [6].

In 2020–2021, while 11% of people living in metropolitan areas accessed Medicare-rebated mental health services, only 6% of those living in remote areas, and 3% of people living in very remote areas accessed equivalent services [10]. A recent scoping review by the Australian Medical Association, however, found that Telehealth services are not being utilised to their full potential in rural and remote areas due to unreliable connectivity [11]. Thus, people in rural and remote areas continue to be disadvantaged in their access to mental health support services.

### Acceptability and suitability of tele-mental health

Although the suitability of tele-mental health for many groups is well documented [12–15], there has been a continued reluctance to engage in telehealth modalities, with face-to-face modalities considered by some to be superior to virtual [12–15]. Rural and remote youth, adults, and the mental health workforce acknowledge tele-mental health is an important complement to face-to-face services [16–18], and many young people with a mental illness living

in rural and remote areas report that tele-mental health is acceptable and equivalent to in-person support [19]. Nevertheless, some still prefer to engage with face-to-face services, and these findings are consistent internationally [16–18, 20, 21]. Benefits of tele-mental health include reducing costs and travel times [16, 17, 19], convenience, shorter wait times [22], and anonymity, particularly for young people identifying as lesbian, gay, bisexual, transgender, queer, intersex, aromantic, asexual and other diverse genders and sexualities (LGBTQIA+) [16]. The perceived impersonal nature of tele-mental health, privacy concerns, and poor audio quality and internet connectivity are concerns for young people, adults, and clinicians delivering tele-mental health services alike [16, 17, 19, 22–25]. A recent systematic review of preferences of rural and remote youth for mental health service access was only able to identify four papers, two Australian, two Canadian, meeting search criteria. The review concluded "Whilst rural and remote youth may prefer to access mental health services in person rather than via tele-health, further well-designed research is needed to better understand under what circumstances this preference holds true and why" [21].

While telehealth has potential to reduce barriers and improve access to healthcare for vulnerable populations, there still remains inequity in telehealth accessibility, for example, those living in rural and remote areas generally have reduced access to reliable technological infrastructure and internet [7, 26, 27]. Some people may lack the requisite skills or equipment [28, 29], and tele-health may be less suitable for some groups. Some people with physical or cognitive disabilities may struggle to operate digital devices, and those with vision or hearing impairments or sensory sensitivities may struggle with digital communication [27, 29, 30], or need to rely on caregiver facilitation within sessions. Evidence on suitability of telehealth for people with autism spectrum disorder is mixed. While some people with autism spectrum disorder may experience challenges building rapport and understanding, and an exacerbation of traits and behaviours during tele-mental health sessions [23, 29], overall, research has demonstrated equal or better outcomes using telehealth compared with face-to-face care for this group [31]. Challenges such as distraction or disengagement have been found with children with disabilities or children who have experienced significant trauma [29,32], and some studies, such as those as outlined in Martiniuk, Toepfer and Lane-Brown's review paper on risks and mitigation strategies for tele-mental health, have reported difficulties assessing and treating people experiencing a mental health crisis such as acute suicidality [33]. Without finding solutions to address these barriers, the Inverse Care Law–where vulnerable and disadvantaged populations require greater access to healthcare, but receive less–will be amplified [34]. More research is needed to determine whether strategies can be developed to improve the suitability of telehealth for these groups.

## Mental health care navigation

Mental health care navigation models have been developed and trialled internationally [35–37]. Early findings are promising, however further research is needed to determine the effectiveness of these models and their various components [37]. Given the known barriers, described above, to accessing face-to-face mental health care for people living in rural and remote areas, an important gap in the existing mental health care navigation literature is whether care navigation can improve the perceived acceptability and suitability of telehealth for help-seekers. This research aims to address this critical gap in knowledge.

## The Navicare model

In 2020 Wesley Research Institute engaged Queensland University of Technology's Australian Centre for Health Services Innovation (AusHSI) to conduct a co-design process to develop a

model of mental health care for the Bowen Basin region of rural and remote Queensland, Australia [4, 5]. Through a series of interviews with service providers and community representatives, and town hall-style workshops open to all interested parties including health consumers, Council, and service providers, the Isaac Navicare model was developed. Care navigation was supported by stakeholders from all groups, however the problem of "navigation to nowhere" was raised because there were insufficient local services to address the needs of the population. A strong theme in the context assessment was the need for the intervention to be local, and therefore stakeholders felt telehealth alone was not the answer. A layered care navigation model was developed, including a supported telehealth site and locally based care navigator. The supported telehealth facility includes a private consultation room with a computer and video camera that clients may use for their telehealth psychology appointments. A care navigator assists clients to access their sessions and can conduct a brief intervention with clients following their appointments if needed. Isaac Navicare commenced a 12-month pilot in November 2021 with a care navigator based at a youth and community centre located in Moranbah. The possibility of stigma related to a person's attendance at a mental health service is minimised by co-locating Navicare with several other community support services and community programs such as a youth group, seniors' activities, and neighbourhood centre program.

## Setting

The Isaac region in Queensland's Bowen Basin consists of predominantly rural, remote, and very remote communities and is characterised by a large mining and agricultural workforce, comprised of both resident and non-resident workers. Although widely-distanced, the many small communities across this region are close-knit. By necessity, the few existing local mental health services work collaboratively across the region to ensure continuity of care for help-seekers. Typically, drive-in-drive-out and fly-in-fly-out workforces are not included in government resource allocations and increase the population of these small local communities by up to 20%, leaving health services under-resourced [38].

Displacement from family and social networks due to workforce requirements and geographic isolation often results in a decline in mental health. Unique risk factors associated with the mining and agricultural industries compound mental health issues, increasing the risk of suicide [39–42]. Compared with other sectors in Queensland, mining and construction employees are 1.72 times more likely to die by suicide [43] and agricultural employees are 2.3 times more likely to die by suicide [40]. Limited access to mental health support services in rural and remote areas likely contributes to the increase in deliberate self-harm and suicides. Notwithstanding these challenges, Isaac locals value the strong community spirit and sense of connectedness within towns and across the broader community.

## Isaac Navicare pilot

Isaac Navicare aims to improve access to timely and clinically appropriate mental health services for people living in the Isaac region who require mental health supports by: (1) Supporting service users to navigate the mental health system by identifying an appropriate and affordable mental health service to meet the user's clinical needs and individual preferences; (2) Reducing waiting time for access to mental health care; (3) Providing a supportive entry point to the mental health system. Engagement with local stakeholders commenced from September 2021. Although the official opening date was November 9, referrals commenced as soon as local providers were aware of the service. Within six months, the high volume of referrals had the service operating at capacity. A second care navigator commenced in December

2022, shortly after the one-year evaluation period ended. Local mental health options are extremely limited, thus it was necessary for the care navigator to find alternative options to avoid "navigation to nowhere."

Relationships were developed with several telehealth and drive-in-drive-out mental health services to enable care navigators to complete referrals directly to their services on behalf of GPs, circumventing the need to send clients back to their doctor to complete the necessary referral paperwork. Furthermore, due to the large client cohort who were unemployed or on a pension or concession of some kind, it was necessary to identify low gap fee and bulk billed counselling and psychology options.

## Aims

This research aimed to: (1) Determine the reach and effectiveness of the Isaac Navicare service in supporting people with mental health concerns to access appropriate mental health services; (2) Identify the model's strengths and limitations; (3) identify adaptations that would improve the acceptability, fit and sustainability of the model.

## Methods

### Design

Ethical approval was granted by UnitingCare Queensland Human Research Ethics Committee (Fisher_20220926), which included the database review, survey and interview. The research design was based on the RE-AIM Framework [44, 45], an implementation science framework focusing on Reach, Effectiveness, Adoption, Implementation and Maintenance. The RE-AIM framework is well suited to the present research as it is a widely used and understood conceptualisation which allows the identification of components of a care delivery model that are effective and those which require adaptation. A strength is a focus on generalisability to other settings and populations [44].

Data were gathered by: (1) Database review; (2) A survey of service users; (3) Interviews with service users. This research related to the pilot phase of the Isaac Navicare service from 9 November 2021 to 8 November 2022, and focuses on the service only, not individual mental health outcomes. Database records for all clients who contacted the service during the pilot phase were included in the study. The service user survey included a modified version of the Navigation Satisfaction Tool (NAVSAT) [46]. To our knowledge, the NAVSAT is the only known validated tool for evaluating service user's satisfaction with care navigation.

The primary outcomes of interest were the reach (number of users, clinical and demographic characteristics) and effectiveness of the service (whether the service user was able to access an appropriate and timely mental health service). Secondary outcomes were adoption, implementation, and adaptations needed to improve sustainability. Queensland University of Technology is conducting a separate study known as The Bridging Study to expand and evaluate Navicare in additional areas of Queensland. This pilot evaluation, led by Wesley Research Institute, was conducted prior to data collection commencing for The Bridging Study and is separate to The Bridging Study led by Queensland University of Technology.

### Participants and recruitment

Isaac Navicare is available to all Isaac area residents and people who work in the Isaac area who are seeking mental health support for themselves or someone else, therefore there were no restrictions on eligibility for the service. Eligible participants were people over the age of 18 who engaged with the Isaac Navicare service during the pilot period to obtain support for

themselves or for a child or family member. Survey and interview data collection was conducted between December 2022 and March 2023. Potential participants were contacted by email in December 2022 and invited to participate in a 15-minute online survey canvassing levels of satisfaction, helpfulness, and effectiveness as well as waiting times. At the completion of the survey, participants were invited to enter contact details for participation in a follow-up interview (S1 File A–Interview Guide). Clients who received a routine check-in call from care navigators were also informed about the study. Interviews were conducted by OF via phone or online meeting according to participant preference. Participants were invited to share further reflections within two weeks of the interview via phone, video conference or email. Exclusion criteria were service users who: were not able to participate in an interview, for example due to health reasons or intellectual impairment; were under 18 years of age; or had previously stated that they did not wish to be contacted. Interviews were conducted between December 2022 and March 2023.

## Analyses

Demographic data were analysed descriptively. Interview transcripts were double coded to the RE-AIM Framework by OF and CG and were analysed using a combined deductive-inductive framework analysis approach [47]. Initial codes were derived from the RE-AIM framework and the research questions, and additional codes were derived inductively. This approach is commonly used in qualitative implementation science research as it allows for a rapid but robust coding process [48]. Trustworthiness was maintained through regular discussion between coders to ensure consistency. Additionally, CL-C acted an external researcher and reliability checker experienced in qualitative analysis, with no role in the development or delivery of the program. CL-C reviewed a sample of four transcripts and the aggregated quantitative data, and in discussion with OF, KM and CG confirmed the overall results. To further guard against researcher bias, qualitative data were triangulated with quantitative data.

## Results

### Participants and recruitment

Of the 197 people who engaged with Isaac Navicare during the pilot, 23 people were not sent the survey due to: age (minor, n = 8); non-consent to be contacted (n = 4); missing contact details (n = 1); intellectual impairment (n = 3); and people who were referred to the service but did not respond to attempts to contact (n = 7). Duplicate emails (n = 11) were removed, e.g., one parent was the contact person for multiple children. In total, 155 emails and 8 text messages were sent in December 2022, with a reminder email/text in January 2023. Eight emails were "undeliverable" so where possible these participants were sent an SMS (n = 7), resulting in 162 invitations. In total, 11 survey responses were received, a response rate of 6.8%. Six survey participants also participated in an interview, with a further five interview participants recruited through routine follow-up by the care navigator. Given the low survey response rate, survey results are not presented, and the findings were only used to confirm and provide context to the database records and interview data. Service user interviews (n = 11) were conducted by phone (n = 10) or Teams (n = 1) between December 2022 and March 2023. Interview duration was between 12 and 41 minutes. All interview participants were referred to Isaac Navicare through general practitioners or school guidance officers. No interview participants took up the offer of sharing a reflection in the two weeks following their interview. All interview participants reported that they had been referred to Isaac Navicare by a local provider such as a general practitioner or a school guidance officer. Five participants had received support for themselves, four as the parent of a child requiring mental health support, and two

participants had received support for both themselves and one or more children. All participants were Bowen Basin residents at the time they were referred to Isaac Navicare.

## Reach

Between 25 October 2021 and 8 November 2022, 197 people (128 adults and 69 minors under 18 years of age) were referred to Isaac Navicare. Demographic characteristics are presented in Table 1. Although the service officially launched on 9 November 2021, service providers began referring clients prior to the launch as soon as they were aware of the service (n = 4). On intake, 48.4% of adult clients were unemployed. Furthermore, 37.5% of adults and 37.7% of minors had a government concession card, i.e., disability, aged, low income, or carer's pension card. Interview participants reported that they felt some groups within the community, particularly young people, were missing out on being able to access mental health supports. There was strong agreement that the Isaac community was largely unaware of Isaac Navicare, limiting the potential client pool.

## Effectiveness

**Supportive entry to mental health system.**   All interview participants agreed that communication with the care navigator was supportive, they felt concerns were heard, and believed they were treated with respect and confidentiality. The empathetic approach of the care navigator and ongoing relational and practical scheduling support were viewed as important. All interview participants stated they would use Isaac Navicare again if needed. Two participants stated that they were unwilling to engage with local providers due to privacy concerns or a previous negative experience with a provider. These participants indicated that Isaac Navicare increased their options.

The extra care, flexibility and responsiveness of the service were also raised:

*I like that. How convenient it was, like I was already on Struggle Street and I couldn't cope with much and the fact that they took care of, you know, the appointments and selecting options for me. And then it was my choice and then she booked me in. I liked the fact that I didn't have to think about it.* (*Service User*)

The care navigator was viewed as providing compassionate and insightful care, with strong understanding of the mental health system. Participants indicated that this engendered trust:

*Yeah, the responsiveness of the service, the ability to feel that they know what they're doing, that they care about you, that they will respond and you're not just left hanging from, you know, being in a sort of stage of oh my goodness, what will I do? That you. . . come off the phone with set actions and what's going to happen.* (*Service User*)

**Suitability of recommended services.**   In total, 276 referrals were made by the care navigator, primarily for mental health treatment (psychology, counselling or social work) as presented in Table 2. Some clients had multiple referrals, the maximum being 13 (Table 2). Conversely, some of the clients in the "did not refer" category had multiple contacts with a care navigator to discuss options, without proceeding to referral, either for mental health or wrap-around support services.

Of the interview participants, 90.9% (n = 10) stated that the service they were referred to was appropriate for their needs, or that they were able to try another service until they found

**Table 1. Demographic characteristics of persons who engaged with Isaac Navicare.**

| | | Total | Adult | Minor (under 18 yrs) |
|---|---|---|---|---|
| **Number of service users** | | 197 | 128 | 69 |
| **Age (years)** | Range | 3 to 73 | 18 to 73 | 3 to 17 |
| | Mean (SD) | 27.9 (±15.8) | 36.4 (±13.5) | 12.6 (±3.5) |
| | Missing | 3 | 3 | 0 |
| **Language spoken at home*** | English | 197 (100%) | 128 (100%) | 69 (100%) |
| | Hindi | 1 (0.5%) | 1 (0.8%) | 0 (0%) |
| **Gender** | Male | 59 (29.9%) | 35 (27.3%) | 24 (34.8%) |
| | Female | 135 (68.5%) | 92 (71.9%) | 43 (62.3%) |
| | Non-binary or other diverse genders | 3 (1.5%) | 1 (0.8%) | 2 (2.9%) |
| **Employment status at intake*** | Mining/ Resources/ Construction | 33 (16.8%) | 33 (25.8%) | 0 (0.0%) |
| | Employed other | 32 (16.2%) | 28 (21.9%) | 4 (5.8%) |
| | Unemployed (Jobseeker/ no income) | 62 (31.5%) | 62 (48.4%) | 0 (0.0%) |
| | NA—Minor | 69 (35.0%) | NA | 69 (100.0%) |
| | NA—School student | 66 (33.5%) | NA | 66 (95.7%) |
| | Missing | 4 (2.0%) | 4 (3.1%) | 0 (0.0%) |
| **Concession card status*** | | | (Own card) | (Parent's card) |
| | Concession card | 74 (37.6%) | 48 (37.5%) | 26 (37.7%) |
| | Department of Veterans' Affairs card | 2 (1.0%) | 2 (1.6%) | 0 (0.0%) |
| | National Disability Insurance Scheme | 8 (4.1%) | 3 (2.3%) | 5 (7.2%) |
| | No concession card | 92 (46.7%) | 54 (42.2%) | 38 (55.1%) |
| | Compulsory Third Party (insurance) claim | 1 (0.5%) | 0 (0.0%) | 1 (1.4%) |
| | Workcover claim | 2 (1.0%) | 2 (1.6%) | 0 (0.0%) |
| | Missing | 29 (14.7%) | 24 (18.8%) | 5 (7.2%) |
| **Location of residence at intake** | Moranbah | 121 (61.4%) | 84 (65.6%) | 37 (53.6%) |
| | Dysart | 25 (12.7%) | 14 (10.9%) | 11 (15.9%) |
| | Middlemount | 9 (4.6%) | 1 (0.8%) | 8 (11.6%) |
| | Nebo | 1 (0.5%) | 1 (0.8%) | 0 (0.0%) |
| | Coppabella | 1 (0.5%) | 1 (0.8%) | 0 (0.0%) |
| | Other Isaac town | 1 (0.5%) | 0 (0.0%) | 1 (1.4%) |
| | Clermont | 18 (9.1%) | 11 (8.6%) | 7 (10.1%) |
| | Mackay Local Government Area | 12 (6.1%) | 10 (7.8%) | 2 (2.9%) |
| | Whitsunday Local Government Area | 1 (0.5%) | 1 (0.8%) | 0 (0.0%) |
| | Central Highlands Local Government Area | 3 (1.5%) | 1 (0.8%) | 2 (2.9%) |
| | Other Local Government Area | 4 (2.0%) | 3 (2.3%) | 1 (1.4%) |
| | No fixed address | 1 (0.5%) | 1 (0.8%) | 0 (0.0%) |
| **Minority Groups*** | No minority group | 110 (55.8%) | 78 (60.9%) | 32 (46.4%) |
| | Disability | 41 (20.8%) | 21 (16.4%) | 20 (29.0%) |
| | Aboriginal | 21 (10.7%) | 16 (12.5%) | 5 (7.2%) |
| | LGBTQIA+ | 11 (5.6%) | 6 (4.7%) | 5 (7.2%) |
| | Both Aboriginal & Torres Strait Islander | 4 (2.0%) | 1 (0.8%) | 3 (4.3%) |
| | Torres Strait Islander | 3 (1.5%) | 1 (0.8%) | 2 (2.9%) |
| | South Sea Islander | 2 (1.0%) | 1 (0.8%) | 1 (1.4%) |
| | English as a second language | 1 (0.5%) | 1 (0.8%) | 0 (0.0%) |
| | Other minority | 5 (2.5%) | 3 (2.3%) | 2 (2.9%) |
| | Missing | 12 (6.1%) | 9 (7.0%) | 3 (4.3%) |

*Total does not equal 100% because clients could select multiple options

one that was appropriate. One participant had not been able to make an appointment. Another participant acknowledged how the rapport between her daughter and the psychologist was hard to establish and so looked for a more suitable match:

*My daughter had a lot of problems, so she can't seem to connect with any psychologist, but [care navigator] was always offering to pick a different one. (Parent)*

**Waiting times.** Most participants stated they were able to arrange a first appointment with a mental health provider quickly, typically within days or weeks. The maximum wait time described by a participant was two months without a response from the provider that they were referred to by the care navigator. Several participants commented on the responsiveness from the care navigator:

*The best thing for me was the compassion and the ease of the service, and how quick it all happened. It wasn't weeks, I wasn't waiting and waiting. Yeah, definitely the ease and compassion and the quickness of everything happening for me. . . I had an appointment within two days. (Service User)*

*She gave me a call. We had to talk and then she said, see what she can do, and then she comes back. Like she always does. And it didn't take long. You wouldn't think a service, you know, on the government thing, you would have to wait. But no, she got a hold of whoever. . . and sorted it out. (Parent)*

*I don't think it was more than a couple of weeks. It was very quick. (Service User)*

**Treatment modality.** Preferred treatment modalities are presented in Table 3. A preference for face-to-face sessions was expressed by 111 (56.3%) clients. Nevertheless, of the 156 clients referred for mental health treatment, most referrals (n = 108, 69.2% of mental health referrals) were made for video or phone telehealth due to a lack of suitable local providers and long waiting lists (Table 3). Of note, 42.6% of clients (n = 84) had access to a free Employee Assistance Program (EAP) psychology or counselling service through their workplace, or parents' workplace for minors, however only 4 clients (4.8% of those with EAP) chose to use the EAP service. Reasons for declining EAP were: insufficient sessions (n = 33); lack of specialists (n = 27); privacy concerns (n = 22); previous experience with EAP was not helpful (n = 14); long travel time (n = 1); long wait time (n = 1); and access to Workcover (n = 1).

Notably, 4.6% of Navicare clients did not receive a referral to any mental health support services and were referred to wrap-around support services only. The people in this group reported that they were experiencing poor mental due to situational stressors. When Navicare completed referrals for the wrap-around support services relevant to each client's needs, the immediate mental health toll was relieved, and these individuals stated that they no longer felt the need to speak with a mental health practitioner.

**Acceptability of telehealth.** The perceived acceptability of telehealth was varied, with some participants reporting an acceptance of telehealth, whether by video or phone, and others feeling strongly that they would only be able to develop therapeutic rapport face-to-face. There were also considerations raised about technology literacy and connectivity in rural locations:

*Service is very hit and miss as well. (Service User)*

The flexibility and confidentiality provided by telehealth proved beneficial, especially in a rural or remote community:

**Table 2. Referrals in and out of Navicare.**

| | | Total | | Adults | | Minors | |
|---|---|---|---|---|---|---|---|
| **Referral source** | General practitioner | 82 | 39.8% | 63 | 47.7 | 19 | 25.7% |
| | Mental health provider | 32 | 15.5% | 17 | 12.9 | 15 | 20.3% |
| | Family member or self-referral | 24 | 11.7% | 13 | 9.8 | 11 | 14.9% |
| | Community support organisation | 21 | 10.2% | 15 | 11.4 | 6 | 8.1% |
| | School staff | 17 | 8.3% | 2 | 1.5 | 15 | 20.3% |
| | Social worker | 12 | 5.8% | 8 | 6.1 | 4 | 5.4% |
| | Alcohol & other drugs service | 7 | 3.4% | 7 | 5.3 | 0 | 0.0% |
| | Allied health provider | 4 | 1.9% | 3 | 2.3 | 1 | 1.4% |
| | Other service provider | 4 | 1.9% | 4 | 3.0 | 0 | 0.0% |
| | National Disability Insurance Scheme provider | 3 | 1.5% | 0 | 0.0 | 3 | 4.1% |
| | Total | 206 | 100.0% | 132 | 100.0% | 74 | 100.0% |
| **Referrals per client** | Did not refer | 32 | 16.2% | 22 | 17.24% | 10 | 14.5% |
| | 1–2 referrals | 143 | 72.6% | 89 | 69.5% | 54 | 78.3% |
| | 3–4 referrals | 13 | 6.6% | 12 | 9.4% | 1 | 1.49% |
| | 5+ referrals | 9 | 4.6% | 5 | 3.97% | 4 | 5.8% |
| | Total | 197 | 100.0% | 128 | 100.0% | 69 | 100.0% |
| **Referrals by service type** | Telehealth psychology or counselling | 103 | 37.3% | 68 | 37.2% | 35 | 37.6% |
| | Face-to-face psychology or counselling (or combination face-to-face & telehealth) | 43 | 15.6% | 18 | 9.8% | 25 | 26.9% |
| | Phone psychology or counselling | 26 | 9.4% | 24 | 13.1% | 2 | 2.2% |
| | Telehealth general practitioner | 17 | 6.2% | 8 | 4.4% | 9 | 9.7% |
| | Social worker | 14 | 5.1% | 0 | 0.0% | 14 | 15.1% |
| | Financial aid or counselling | 11 | 4.0% | 10 | 5.5% | 1 | 1.1% |
| | Centrelink agent | 9 | 3.3% | 8 | 4.4% | 1 | 1.1% |
| | Local general practitioner | 9 | 3.3% | 8 | 4.4% | 1 | 1.1% |
| | Housing | 8 | 2.9% | 8 | 4.4% | 0 | 0.0% |
| | Community or social group | 7 | 2.5% | 6 | 3.3% | 1 | 1.1% |
| | Allied health | 6 | 2.2% | 5 | 2.7% | 1 | 1.1% |
| | Free legal service | 6 | 2.2% | 6 | 3.3% | 0 | 0.0% |
| | Alcohol & other drugs service | 6 | 2.2% | 6 | 3.3% | 0 | 0.0% |
| | Other provider | 4 | 1.4% | 2 | 1.1% | 2 | 2.2% |
| | Emergency services | 3 | 1.1% | 2 | 1.1% | 1 | 1.1% |
| | National Disability Insurance Scheme support | 2 | 0.7% | 2 | 1.1% | 0 | 0.0% |
| | Employment service | 2 | 0.7% | 2 | 1.1% | 0 | 0.0% |
| | Total referrals to mental health and wrap-around support services | 276 | 100.0% | 183 | 100.0% | 93 | 100.0% |

> *I was very paranoid about people knowing that I was parked out front of a service that you know was for mental health or for counselling. . . That really did hold me back at times in getting counselling because I did want help, but I didn't want to do that in our small community.* (Service User)

Advantages included mobility, with one participant describing going on extended holiday and still being able to attend psychology appointments by phone, as well as being able to log in from a comfortable, familiar, private space:

> *Yeah, I probably found it better because I didn't really have distractions or thoughts about you know, being judged or anything like that, it was a bit it was a bit of a nicer environment, if I'm being honest.* (Service User)

**Table 3. Mental health treatment modality–preferred versus referral made.**

| | TOTAL | | ADULTS | | MINORS | |
|---|---|---|---|---|---|---|
| | **Preference** | **Referred to (/197)** | **Preference** | **Referred to (/128)** | **Preference** | **Referred to (/69)** |
| Face-to-face | 111 (56.3%) | 45 (22.8%) | 52 (40.6%) | 18 (14.1%) | 59 (85.5%) | 27 (39.1%) |
| Video telehealth | 14 (7.1%) | 90 (45.7%) | 14 (10.9%) | 63 (49.2%) | 0 (0.0%) | 27 (39.1%) |
| Phone telehealth | 2 (1.0%) | 13 (6.6%) | 2 (1.6%) | 13 (10.2%) | 0 (0.0%) | 0 (0.0%) |
| Supported video telehealth | 3 (1.5%) | 5 (2.5%) | 3 (2.3%) | 5 (3.9%) | 0 (0.0%) | 0 (0.0%) |
| Combination (face-to-face and telehealth) | 0 (0.0%) | 3 (1.5%) | 0 (0.0%) | 1 (0.8%) | 0 (0.0%) | 2 (2.9%) |
| No preference | 36 (18.3%) | | 31 (24.2%) | | 5 (7.2%) | |
| Preference unknown | 31 (15.7%) | | 26 (20.3%) | | 5 (7.2%) | |
| No mental health referral made | | 41 (20.8%) | | 28 (21.9%) | | 13 (18.8%) |
| Total (clients) | 197 | 156 (79.2% clients referred) | 128 | 100 (78.1% adults referred) | 69 | 56 (81.2% minors referred) |

Note: Total referrals does not equal 100% because some clients were referred to more than one service, or were not referred to any service

Telehealth was thus seen as an acceptable option to limit exposure and stigma in the community. Participants described a growing acceptance of telehealth during and since the COVID-19 pandemic. However, assistance for people who struggle with technology was raised:

> *But I mean, I think of people I've known, my parents, my dad, if he had to be connected with telehealth or a service of that sort through a computer, he would definitely need somebody to be able to help him do that. Um, you know, and it's sort of that era and that generation that may need some help. And then also those people who don't have the internet at home.* (*Service User*)

Participants reported that they were more likely to trial telehealth because they knew they could change to another service if they did not find it effective. This was a key enabler to the uptake of tele-mental health services during the pilot.

> *[Care navigator] made me feel comfortable that if I wasn't going to work well with a telehealth type appointment like video calls or with these particular counsellors we could try another service. We could try another person. It just made me feel as though it's ok to say this isn't working for me.* (*Service User*)

**Young people and telehealth.**   Five participants raised additional considerations about the suitability of telehealth for children and teenagers. One concern was privacy within the home, especially when using a shared family computer:

> *Possibly, like [name] might be all right here talking, but then is he going to open up as much as if he was in a room somewhere where we weren't lingering in the background.* (*Parent*)

Another limitation noted was the ability of clients to walk away or turn off the computer:

> *The face to face, I feel, especially for my son having ADHD. He's very much it has to be in person. . .Or you will not get anything out of him. We've had telehealth appointments with his*

*paediatrician who's in Brisbane and there's, there's nothing. He just he walks out, he walks off and leaves and goes I'm bored. I don't want to talk to that person. Because he's very much. . . a face-to-face person you have to be hands on with him.* (*Parent*)

Some participants were able to make adaptations and improve the acceptability of telehealth:

*My daughter struggles with that because she had body dysmorphia and she really struggled. . . she couldn't look at the screen. . . She didn't like it, but she adapted, so like I said, she would have me with her, and the focus would be more looking at me, but talking to her so she could then sort of look at me and avoid the whole camera thing. So she did quite well doing it that way.* (*Parent*)

## Adoption

In total, 206 referrals were received for 197 clients as presented in Table 2. Nine clients were referred to Navicare by two different service providers. In each instance one was from a GP. Only 24 (11.7%) referrals were self/family or website referrals, and the remaining 88.3% were from local health or community service providers or schools.

## Implementation

Having a local navigator was regarded as important by interview participants, even though some had not met the navigator face-to-face. Participants particularly valued the care navigator's detailed knowledge of local providers and challenges. Interview participants were not aware of the supported tele-health site. Nevertheless, five clients were referred to this service. Ten interview participants said the site was a good idea and they could think of people who would use it. They had not used it themselves due to distance, appropriate set up at home, or ability to access local face-to-face services. Some participants could not identify the difference between the care navigation service and mental health service providers, with some believing them to be one and the same.

## Maintenance: Adaptations and sustainability

Interview participants described a lack of awareness of Isaac Navicare in the local community, and recommended promotion of the service directly to the community, not just to service providers. Participants felt strongly that the service needed to be expanded to other townships in the Bowen Basin region to increase accessibility. Further, the service was quickly operating at capacity which indicates the need for and success of the pilot but also the need for more care navigators.

# Discussion

In rural and remote areas there is both greater need for, and less access to, suitable mental health services compared to urban centres. Isaac Navicare aims to improve access to timely and clinically appropriate mental health services for people living in the Isaac region of Queensland, Australia, in large part by improving the acceptability and uptake of telehealth services. The present study used the RE-AIM implementation science framework to evaluate a one-year pilot of this service. Overall, the service was perceived by users as supportive, appropriate for their needs, and resulted in timely appointments with mental health care providers for those who needed them. Although many service users initially expressed reluctance to

engage in tele-mental health, the lack of suitable local providers, combined with tailored navigation from a knowledgeable local care navigator, resulted in increased willingness of service users to trial tele-mental health, with many finding it appropriate for their needs. This pragmatic, technologically mediated solution to the dearth of local providers was crucial to meeting the aim of reducing waiting times to access mental health services.

The model has several strengths. For clients who required several referrals to find acceptable and appropriate support, the care navigator provided a sense of continuity of care. Given well-recognised challenges with mental health help-seeking and risks of disengagement, [49] this is a noteworthy benefit. Approximately 80% of clients received a referral to a mental health service with most interview participants indicating that they found acceptable and appropriate support.

The service was also successful in reaching some disadvantaged groups with approximately half of adult clients being unemployed and over a third of clients holding a government concession card–groups who typically face significant barriers to obtaining care [44]. Given that the proportion of the Queensland population who identify as being Aboriginal and/or Torres Strait Islander is 4.6% [50], it is noteworthy that 14.2% of Isaac Navicare service users identified as either Aboriginal and/or Torres Strait Islander. Although the proportion of First Nations Queenslanders with a mental illness is higher than the total Queensland population–12.5% versus 9.6% respectively [51], the proportion of First Nations Isaac Navicare service users was higher than anticipated by the research team, based on these population characteristics. It is possible that this could indicate a latent need for mental health services in the local First Nations population, and/or possibly a high level of acceptance of the service by local First Nations communities. Future research to further understand these dynamics would be warranted to identify whether Isaac Navicare, and care navigation more broadly, are effective options for increasing equitable access to mental health services for First Nations communities.

This evaluation also found that 4.6% of clients who were referred to Navicare for mental health support only required referral for wrap-around supports related to situational stressors. Once the immediate stress was relieved by referrals for wrap-around supports relevant to their situation, these individuals reported that they no longer needed to speak with a mental health clinician. This prevented unnecessary referrals to mental health support services, freeing up appointments for people who required clinical support. On a larger scale, this tailored support may reduce nonessential referrals, alleviate pressure on our already stretched MH system, and translate to significant cost savings.

Despite the clear strengths of the service, these data suggest the service had some limitations or scope for adaptation. It is likely that the service did not meet the needs of all people requiring mental health services in the Isaac region. The service was busy from inception, with inward referrals from general practitioners and the local acute mental health service starting prior to commencement. There was, therefore, a need for more care navigators to provide greater capacity. It is also likely that people with the greatest disadvantage who were not engaged with any local services such as a general practitioner or school guidance officer, were not aware of the service. This is consistent with the low up-take of the assisted tele-health facilities. The service did not reach those for whom basic digital access was an issue. Key adaptations, therefore, would include expanding the capacity of the service, direct outreach to smaller communities in this geographically large region, and better outreach to people more likely to use the supported telehealth service due to limited technological access or knowledge.

This study contributes to the growing body of literature around the effectiveness of mental health care navigation and addresses a critical gap in knowledge about whether care navigation can improve the uptake and acceptability of tele-mental health for people living in rural and

remote areas. Previous studies such as those outlined in Waid, Halpin and Donaldson's 2021 review paper [37] have identified common features of care navigation services that were also present in Isaac Navicare model, such as brief assessment and triage, referral and follow-up support. The findings of this study support the effectiveness of these model components and their inclusion in future care navigation services.

The pragmatic implementation science framework analysis approach used enabled the authors to quickly analyse and triangulate the quantitative and qualitative data and relate them to a pre-determined set of outcomes derived from the RE-AIM Framework, complemented by implementation science theory and inductively derived codes. This approach successfully enabled us to review and analyse these data in a timely manner so feedback could be provided to service providers. Although the authors did not keep a record of analysis time, we noted anecdotally that this coding strategy substantially reduced the analysis time compared with traditional qualitative methods we have used previously, while still resulting in robust and analysis and interpretation of these data.

The research also has strengths and weaknesses. A strength is the substantial amount of qualitative data gathered in interviews with clients. Clear and consistent results were identified from these data. Conversely, little data was gathered by survey due to low response rates. While the low response rate is not surprising in this context it remains a limitation.

## Conclusion

The high uptake of the Isaac Navicare service, even prior to formal commencement of the pilot, highlights the unmet need for mental health services in rural areas. Coupling tele-mental health with local care navigators led to increased uptake of telehealth services, short waiting times for help-seekers, referrals to suitable and appropriate mental health and complementary services, and high service user satisfaction. Important adaptations were identified to improve the effectiveness of the service and increase uptake, although it is acknowledged that additional staffing will be required to manage any increase in referrals to the service. Future cost-benefit analysis would be warranted. Given the effectiveness of the Isaac Navicare model in increasing access to mental health services for people living in rural areas, even in this brief pilot phase, it may be a suitable model for expansion to or replication in other areas. This study demonstrates that telehealth alone is not enough to enable people living in rural and remote areas to access appropriate mental health services, therefore policy-makers need to consider complimentary services such as care navigation to improve timely and equitable mental healthcare.

## Supporting information

**S1 File.**
(DOCX)

## Acknowledgments

The authors acknowledge the contribution of people with lived experience of mental illness who helped to design the Isaac Navicare service, who accessed the service and provided valuable feedback as part of this evaluation, and who have promoted the service within the Bowen Basin community. Sincere thanks also go to their families, local stakeholders such as service providers and government, and community members who contributed to both the design of this service, and this evaluation. We would like to acknowledge the important contribution of Dr Bridget Abell, Research Fellow, AusHSI, QUT, who led the original co-design study to develop the Isaac Navicare model, and who collaborated with the research team on the design

of this evaluation. Thanks also go to Keeley Ryan, Care Navigator, Isaac Navicare, who supported the participant recruitment process, and Belinda Moshi who assisted with formatting the final manuscript.

## Author Contributions

**Conceptualization:** Olivia J. Fisher, Kelly McGrath.

**Data curation:** Olivia J. Fisher, Kelly McGrath.

**Formal analysis:** Olivia J. Fisher, Kelly McGrath, Caroline Grogan, Wendell Cockshaw, Chez Leggatt-Cook.

**Methodology:** Olivia J. Fisher.

**Project administration:** Olivia J. Fisher.

**Writing – original draft:** Olivia J. Fisher, Kelly McGrath, Caroline Grogan, Wendell Cockshaw, Chez Leggatt-Cook.

**Writing – review & editing:** Olivia J. Fisher, Kelly McGrath, Caroline Grogan, Wendell Cockshaw, Chez Leggatt-Cook.

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
