## [Decision Letter · Decision Letter 0]

13 Nov 2023

PONE-D-23-24611Care navigation leads to improved acceptability and uptake of tele-mental health in rural and remote communitiesPLOS ONE

Dear Dr. Fisher,

Thank you for submitting your manuscript to PLOS ONE. After careful consideration, we feel that it has merit but does not fully meet PLOS ONE’s publication criteria as it currently stands. Therefore, we invite you to submit a revised version of the manuscript that addresses the points raised during the review process.

We look forward to receiving your revised manuscript.

Kind regards,

Sonu Bhaskar, MD PhD

Academic Editor

PLOS ONE

Journal Requirements:

2. Thank you for providing the following Funding Statement: 

“The authors OF, CG and KM acknowledge a potential conflict of interest due to their roles in the design (OF) or service delivery (KM and CG) of Isaac Navicare. To minimise any impact of this potential bias, CL-C was invited to be a member of the research team external to the project with no conflict of interest to declare. CL-C reviewed a selection of transcripts along with the quantitative data to ensure the authenticity of the results presented, and the accurate interpretation of these data. CL-C also contributed to the development of the manuscript.”

We note that one or more of the authors is affiliated with the funding organization, indicating the funder may have had some role in the design, data collection, analysis or preparation of your manuscript for publication; in other words, the funder played an indirect role through the participation of the co-authors.

If the funding organization did not play a role in the study design, data collection and analysis, decision to publish, or preparation of the manuscript and only provided financial support in the form of authors' salaries and/or research materials, please review your statements relating to the author contributions, and ensure you have specifically and accurately indicated the role(s) that these authors had in your study in the Author Contributions section of the online submission form. Please make any necessary amendments directly within this section of the online submission form.  Please also update your Funding Statement to include the following statement: “The funder provided support in the form of salaries for authors [insert relevant initials], but did not have any additional role in the study design, data collection and analysis, decision to publish, or preparation of the manuscript. The specific roles of these authors are articulated in the ‘author contributions’ section.”

If the funding organization did have an additional role, please state and explain that role within your Funding Statement.

Please also provide an updated Competing Interests Statement declaring this commercial affiliation along with any other relevant declarations relating to employment, consultancy, patents, products in development, or marketed products, etc. 

Additional Editor Comments (if provided):

Thanks for submitting your work to PLOS One. Based on comments from the reviewers, I would like to invite you to revise your manuscript, and provide a point-by-point rebuttal the the comments provided.

Reviewers' comments:

Reviewer's Responses to Questions

**Comments to the Author**

1. Is the manuscript technically sound, and do the data support the conclusions?

Reviewer #1: Yes

Reviewer #2: Yes

2. Has the statistical analysis been performed appropriately and rigorously? 

Reviewer #1: Yes

Reviewer #2: Yes

3. Have the authors made all data underlying the findings in their manuscript fully available?

Reviewer #1: No

Reviewer #2: No

4. Is the manuscript presented in an intelligible fashion and written in standard English?

Reviewer #1: Yes

Reviewer #2: Yes

5. Review Comments to the Author

Reviewer #1: Care navigation leads to improved acceptability and uptake of tele-mental health in rural and remote communities

The authors did a great job given that telehealth is becoming an innovative approach for tele-mental health. Generally, the manuscript is technically sound well written. However, they need to address these minor comments below to help strengthen the paper.

Study Title

• The study title needs to include the geographical location (country) of the study.

Abstract

• Navicare need to be defined or explained in the introduction.

• The authors need to define the evidence gap and the study aim in the introduction.

Methods

• The authors need define the study participants.

• The authors need to also indicate the end date.

Discussion

• The authors should consider deleting discussion as a heading and integrating the information to the conclusion as policy recommendation/s.

Introduction

• The phrase after the reference at line 77 also need referencing.

• The sentence at line 87 need to be referenced.

• The sentence at line 97 need to be referenced.

• The sentence at line 101 need to be referenced.

• LGBTQIA+ at line 103 need to be written in full on first use and put in brackets (LGBTQIA+)

• The sentence at line 109 ending with via telehealth, need to be referenced.

• The sentence at line 112 need to be referenced

• At line 125, authors mentioned studies as evidence but referenced only one study (36).

Reviewer #2: This article presents an evaluation study of a pilot model of mental health care, co-designed with stakeholders, that was implemented in a regional/rural/remote setting in Queensland, Australia. The evaluation uses the RE-AIM framework and draws on existing methods to incorporate deductive and inductive qualitative analysis and triangulation. An independent researcher was included in the analysis process to help overcome researcher bias. The evaluation methods are appropriate and described well. The findings are reported in appropriate detail and the discussion is supported by the reported findings. Overall, the study found that the mental health care navigation service was highly acceptable to consumers and increased access to mental health care. Mental health care in rural and remote Australia is challenging and this study makes an important contribution to this field.

There are a few areas that I suggest should be addressed to improve clarity for the reader and strengthen the paper:

1. The introduction paragraph contains helpful information for context however should be revised slightly for an international audience. i.e. it is not clear that this paragraph is focused on Australia.

2. Line 132 – the Bowen Basin region is first introduced here – it would be helpful for the reader to give an indication of where this is.

3. Line 139 – Please clarify what is meant by a supported telehealth site – it is mentioned in a few areas of the paper but it is unclear for the reader what this is.

4. Line 140 – what is meant by ‘non-stigmatising’ location?

5. Line 144 – Please clarify if the agricultural workforce are resident or non-resident.

6. Line 186 – The RE-AIM Framework citation is missing.

7. Line 191 – Was ethic approval needed/obtained for use of the database in the evaluation?

8. Line 201 – What is the Bridging Study? This seems like something separate from this evaluation study – please clarify or remove the statement.

9. Line 203 – Consider using the heading ‘Recruitment’ or similar

10. Line 204 is repeated in Line 213

11. Line 214 – Did anyone take up the option to share a reflection? I don’t think I saw this in the findings and will be useful for the reader to include in the findings section and understand how this was incorporated into the analysis, if relevant.

12. Line 244 – a brief overview of interview participant characteristics/demographics would be useful for the reader.

13. The quotes are well chosen and convey the findings well. A descriptor at the end of each quote would further strengthen the use of quotes – e.g. parent

14. Line 278 – the 276 referrals relate to referrals made by the care navigator. Please include this information so the reader can easily distinguish this from referring providers to the Navicare service.

15. Table 2 – I note that this table has a total of 32 ‘did not refer’. But in Table 3, the count is 41. Is there an error?

16. Line 291 – Is there a maximum number of weeks that could be added here?

17. Line 400 – At least 11% of clients identified as First Nations – this is important and I suggest that the authors consider inclusion of this as a discussion point.

18. Line 431 – Please consider - ‘reduced’ waiting times or ‘short’ waiting times?

19. The article describes a rigorous evaluation study of a model of care that has increased access to mental health care. Engagement with the existing literature about similar care navigation models for mental health would further strengthen this article. What does this study add? What are the key learnings?

6. PLOS authors have the option to publish the peer review history of their article (what does this mean?). If published, this will include your full peer review and any attached files.

Reviewer #1: **Yes: **Alexander Laar

Reviewer #2: No

---

## [Author Response · Author response to Decision Letter 0]

30 Nov 2023

A detailed response to all reviewer comments has been uploaded as a separate file.

---

## [Decision Letter · Decision Letter 1]

30 Jan 2024

Care navigation addresses issues of tele-mental health acceptability and uptake in rural and remote Australian communities

PONE-D-23-24611R1

Dear Dr. Fisher,

We’re pleased to inform you that your manuscript has been judged scientifically suitable for publication and will be formally accepted for publication once it meets all outstanding technical requirements.

Kind regards,

Sarah Kathrine Day, PhD

Academic Editor

PLOS ONE

Reviewers' comments:

Reviewer's Responses to Questions

**Comments to the Author**

1. If the authors have adequately addressed your comments raised in a previous round of review and you feel that this manuscript is now acceptable for publication, you may indicate that here to bypass the “Comments to the Author” section, enter your conflict of interest statement in the “Confidential to Editor” section, and submit your "Accept" recommendation.

Reviewer #1: All comments have been addressed

Reviewer #2: All comments have been addressed

2. Is the manuscript technically sound, and do the data support the conclusions?

Reviewer #1: Yes

Reviewer #2: Yes

3. Has the statistical analysis been performed appropriately and rigorously? 

Reviewer #1: Yes

Reviewer #2: Yes

4. Have the authors made all data underlying the findings in their manuscript fully available?

Reviewer #1: Yes

Reviewer #2: Yes

5. Is the manuscript presented in an intelligible fashion and written in standard English?

Reviewer #1: Yes

Reviewer #2: Yes

6. Review Comments to the Author

Reviewer #1: The authors have addressed all my review comments. They have also adhered to the research or publication ethics of the journal.

Reviewer #2: (No Response)

7. PLOS authors have the option to publish the peer review history of their article (what does this mean?). If published, this will include your full peer review and any attached files.

Reviewer #1: **Yes: **Alexander Laar

Reviewer #2: No

---

## [Editor Report · Acceptance letter]

26 Mar 2024

PONE-D-23-24611R1 

PLOS ONE

Dear Dr. Fisher, 

I'm pleased to inform you that your manuscript has been deemed suitable for publication in PLOS ONE. Congratulations! Your manuscript is now being handed over to our production team.

Kind regards, 

on behalf of

Dr. Sarah Kathrine Day 

Academic Editor

PLOS ONE